# The Mediating Effect of E-Cigarette Harm Perception in the Relationship between E-Cigarette Advertising Exposure and E-Cigarette Use

**DOI:** 10.3390/ijerph19106215

**Published:** 2022-05-20

**Authors:** Nan Jiang, Shu Xu, Le Li, Omar El-Shahawy, Nicholas Freudenberg, Jenni A. Shearston, Scott E. Sherman

**Affiliations:** 1Department of Population Health, Grossman School of Medicine, New York University, New York, NY 10016, USA; le.li@nyulangone.org (L.L.); omar.elshahawy@nyulangone.org (O.E.-S.); scott.sherman@nyulangone.org (S.E.S.); 2School of Global Public Health, New York University, New York, NY 10003, USA; sx5@nyu.edu (S.X.);; 3School of Public Health, City University of New York, New York, NY 10027, USA; nick.freudenberg@sph.cuny.edu; 4Mailman School of Public Health, Columbia University, New York, NY 10032, USA; js5431@cumc.columbia.edu; 5Department of Medicine, VA New York Harbor Healthcare System, New York, NY 10010, USA

**Keywords:** e-cigarette, smoking, advertising, perception, mediator, college student

## Abstract

Exposure to e-cigarette advertising is associated with e-cigarette use among young people. This study examined the mediating effect of e-cigarette harm perception on the above relationship. Cross-sectional survey data were collected from 2112 college students in New York City in 2017–2018. The analytic sample comprised 2078 participants (58.6% females) who provided completed data. Structural equal modeling was performed to examine if e-cigarette harm perception mediated the relationship between e-cigarette advertising exposure (via TV, radio, large signs, print media, and online) and ever e-cigarette use and susceptibility to e-cigarette use. About 17.1% of participants reported ever e-cigarette use. Of never users, 17.5% were susceptible to e-cigarette use. E-cigarette advertising exposure was mainly through online sources (31.5%). Most participants (59.4%) perceived e-cigarettes as equally or more harmful than cigarettes. Advertising exposure showed different effects on e-cigarette harm perception depending on the source of the advertising exposure, but perceiving e-cigarettes as less harmful than cigarettes was consistently associated with e-cigarette use and susceptibility. Low harm perception mediated the association between advertising exposure (via online, TV, and radio) and ever e-cigarette use and between online advertising exposure and e-cigarette use susceptibility. Regulatory actions are needed to address e-cigarette marketing, particularly on the Internet.

## 1. Introduction

Young adults are vulnerable to tobacco use and nicotine dependence [1,2]. Data from the National Health Interview Surveys showed that during 2014 and 2018, current (past 30-day) e-cigarette use remained highest among young adults (aged 18–24) and the rates increased over the years among this population (5.1% to 7.6%), whereas the rates remained stable or declined among adults aged 25 years and older [3].

The rise in e-cigarette use can be attributed to multiple factors, including the aggressive marketing targeting young people [4,5]. E-cigarettes are often promoted as healthier alternatives to traditional cigarettes, smoking cessation aids, and socially acceptable products [5,6,7,8]. E-cigarettes have been widely marketed through various sources, including retail stores, the Internet, social media sites, TV and radio, newspapers and magazines, posters and billboards, and public events, such as festivals [9]. 

A significant proportion of young adults are exposed to e-cigarette advertising [10,11]. From 2011 to 2013, exposure to TV e-cigarette advertising increased 321% among young adults [12]. E-cigarette advertising exposure may impact the uptake of e-cigarettes. Longitudinal data have illustrated that e-cigarette advertising exposure predicts subsequent e-cigarette use among tobacco-naïve young adults [10]. In a randomized controlled trial, brief e-cigarette advertising exposure was found to be associated with increased experimentation with e-cigarettes and greater susceptibility to e-cigarette use in young adults [13]. 

The underlying mechanism of how advertising exposure affects people’s e-cigarette use behaviors has not been fully explored. A potential mechanism is that exposure to e-cigarette advertising reduces the degree of harm perception of e-cigarettes, which, in turn, stimulates the uptake of e-cigarettes. The association between e-cigarette advertising exposure and the low harm perception of e-cigarettes has been found among young adults [14,15,16], and the harm belief is known to play an important role in people’s behavior [17]. For US adults, a commonly cited reason for e-cigarette use is the low harm perception of e-cigarettes [18,19]. Longitudinal and cross-sectional data have revealed that the low relative harm perception of e-cigarettes compared to cigarettes is associated with e-cigarette initiation, ever and current e-cigarette use, and greater susceptibility to e-cigarette use [16,20,21,22,23,24,25]. The purpose of this study is to examine whether the harm perception of e-cigarettes mediates the association between advertising exposure and e-cigarette use and susceptibility in college students. Understanding the mechanism of how advertising exposure affects e-cigarette use is fundamental to develop prevention and intervention efforts that tackle the rise in e-cigarette use.

## 2. Materials and Methods

### 2.1. Survey Procedures and Participants

Between September 2017 and February 2018, students from a large public university in New York City completed a Tobacco Use and Health Behavior Survey (hereafter referred to as the “Survey”), a survey designed to assess a broad range of health-related behaviors among college students. An invitation email that described the nature of the study with a link to the online survey was sent to 5300 students who were randomly selected across all 25 campuses of the university. Students who did not complete the online survey were invited to complete the Survey via the phone. It took roughly 20 min to complete the Survey. Informed consent was obtained from study participants. Each participant was compensated with a $20 Amazon gift card. A stratified probability sampling method was used to generate a representative sample of students. Further details regarding the study are published elsewhere [26].

A total of 2112 students completed the Survey, yielding an approximate response rate of 40%. The analytic sample included 2078 students who provided completed data for the demographics, e-cigarette advertising exposure, harm perception, and tobacco use. The study protocol was approved by the Institutional Review Boards of the New York University Grossman School of Medicine and the City University of New York.

### 2.2. Measures

#### 2.2.1. Demographic Characteristics

Participants reported age, gender, race/ethnicity, and native status (US vs. foreign-born). Age was collapsed into 4 groups (i.e., 18–20, 21–24, 25–34, and ≥35 years), and race/ethnicity was divided into 5 categories (i.e., non-Hispanic White, non-Hispanic Black, non-Hispanic Asian, Hispanic, and other).

#### 2.2.2. E-Cigarette Advertising Exposure

Following prior research [27], we assessed advertising exposure using one question, “In the last 6 months, have you noticed advertising in any of the following places for e-cigarettes? On television; On the radio or Internet radio (e.g., Pandora and Spotify); On large signs (e.g., bus shelters, subway platforms, trains, and top of taxis); In print media (e.g., newspapers and magazines); Online.” Dichotomous response choices included “Yes” and “No” for each of the 5 sources.

#### 2.2.3. Tobacco Use

Two items assessed tobacco use status: “Which, if any, of the following tobacco or nicotine products have you ever used or tried, even one puff?” and “Which of the following products have you used in the past 30 days?” Response options included “cigarettes”, “cigars”, “hookah”, “e-cigarettes”, “little cigars/cigarillos”, “chewing tobacco, snuff or dip”, “other tobacco product”, and “I have never used any tobacco or nicotine products”. Responses to the two questions established categories for cigarette smoking, alternative tobacco use, and e-cigarette use. Cigarette smoking status included never (those who did not select “cigarettes” in either question), former (those who selected “cigarettes” in the first question but not the second one), and current smoking (those who selected “cigarettes” in both questions). Similarly, alternative tobacco use was categorized as never use (for those who selected “I have never used any tobacco or nicotine products” or selected only “cigarettes” or “e-cigarettes” in the first question), former use (for those who selected any tobacco products excluding cigarettes and e-cigarettes in the first question but not the second one), and current use (for those who selected any tobacco products excluding cigarettes and e-cigarettes in both items). E-cigarette use status was categorized as never (those who did not select “e-cigarettes” in either question), ever (those who selected “e-cigarettes” in the first question only), and current use (those who selected “e-cigarettes” in both questions). 

#### 2.2.4. Susceptibility to E-Cigarette Use

Measures of susceptibility to e-cigarette use were adapted from established measures of susceptibility to cigarette smoking [28]. Participants were asked three items: “Have you ever been curious about using e-cigarettes?” “Do you think that you will try an e-cigarette in the next year?” “If one of your friends or somebody close to you offered you an e-cigarette, would you try it?” Response choices included “definitely yes”, “probably yes”, “probably not”, and “definitely not”. E-cigarette never users who answered “definitely not” to all of the three items were considered to be unsusceptible to e-cigarette use and otherwise, they were considered susceptible.

#### 2.2.5. Harm Perception of E-Cigarettes

We assessed the relative harm perception of e-cigarettes by asking, “Compared to cigarettes, how harmful do you think e-cigarettes are to a person’s health?” Answer options included “a lot less harmful”, “a little less harmful”, “about the same”, “a little more harmful”, and “a lot more harmful”. Following prior research [23,29], we collapsed responses into two groups (i.e., “equally/more harmful” and “less harmful”).

### 2.3. Statistical Analyses 

We conducted analyses using the R lavaan package [30]. All analyses were weighted to account for the complex sampling design. Descriptive statistics summarized participants’ demographic and tobacco use characteristics, e-cigarette advertising exposure, and harm perception. 

We then conducted structural equal modeling to examine the mediating effect of e-cigarette harm perception (mediator) on the relationship between e-cigarette advertising exposure and two outcomes, including ever e-cigarette use and susceptibility to e-cigarette use. Current e-cigarette use was not included in our analyses as an outcome because of its low frequency (*n* = 73; 3.6% of the sample), preventing us from performing probit regression models. Specifically, we first performed separate probit regression models to examine the association between advertising exposure (through each of the 5 sources) and the mediator (path α in Figure 1a), controlling for covariates (i.e., demographics, cigarette smoking status, and alternative tobacco use). Second, we conducted separate probit regression models to assess the association between the mediator and outcomes (path β in Figure 1a), adjusting for advertising exposure and covariates. In these regression models, coefficients of advertising exposure accounted for direct effect of advertising exposure on outcomes (path c’ in Figure 1a), adjusting for mediators and covariates. We also performed probit regression models to assess the association between advertising exposure and outcomes adjusting for covariates (path c in Figure 1b), and the coefficients accounted for the total effect of advertising exposure on outcomes. Last, we calculated the product of coefficients α and β, which accounted for the indirect effect of advertising exposure on outcomes through the mediator. Bootstrapping with 1000 replications was used to estimate the 95% confidence interval (CI) of the indirect effect [31]. If the 95% CI did not include the null, we would conclude that the association between e-cigarette advertising exposure and outcomes was mediated by e-cigarette harm perception. For interpretation purposes, we converted probit regression coefficients to adjusted odds ratios (AORs) by exponentiating 1.8 times the values of the coefficients [32], and then we reported the AORs and 95% CIs.

## 3. Results

### 3.1. Demographic and Tobacco Use Characteristics

Participants were an average of 25.2 years (95% CI: 24.9, 25.6) with 58.6% being females (Table 1). One-third (33.8%) of participants were Hispanic and 62.3% were born in the US. Participants included 19.4% former cigarette smokers, 6.9% current cigarette smokers, 26.3% former users of alternative tobacco products, and 7.8% current alternative tobacco users. About 17.1% of participants reported ever e-cigarette use, including 3.6% of current users. Of those who had never used an e-cigarette (*n* = 1727), 17.5% reported susceptibility to e-cigarette use. Most participants (59.4%) perceived e-cigarettes as equally or more harmful than cigarettes. Past 6-month exposure to e-cigarette advertising was mainly through online sources (31.5%), followed by TV (18.8%), large signs (17.9%), print media (16.1%), and radio (8.1%). 

### 3.2. Relationship between E-Cigarette Advertising Exposure and Harm Perception and E-Cigarette Use Outcomes 

Participants who reported exposure to online e-cigarette advertising were more likely to report a low relative harm perception of e-cigarettes compared to cigarettes (AOR = 1.29, 95% CI: 1.04, 1.60; Table 2); the low harm perception was associated with higher adjusted odds of ever e-cigarette use (AOR = 1.75, 95% CI: 1.44, 2.18). Similarly, among e-cigarette never users, online exposure was associated with a low harm perception of e-cigarettes (AOR = 1.31, 95% CI: 1.04, 1.72), which was associated with higher adjusted odds of e-cigarette use susceptibility (AOR = 1.11, 95% CI: 1.08, 1.16). Bootstrap 95% CIs derived from 1000 samples illustrated significant indirect effects of online advertising exposure on ever e-cigarette use (AOR = 1.08, 95% CI = 1.02, 1.18) and e-cigarette use susceptibility (AOR = 1.02, 95% CI = 1.01, 1.04). Findings indicated that the low harm perception of e-cigarettes mediated the relationship between online advertising exposure and the two e-cigarette use outcomes. 

A converse relationship was observed between advertising exposure via TV or radio and e-cigarette harm perception. Participants who reported TV or radio exposure were less likely to perceive e-cigarettes as less harmful than cigarettes (TV: AOR = 0.70, 95% CI: 0.52, 0.91; radio: AOR = 0.65, 95% CI: 0.43, 0.95). However, the low relative harm perception of e-cigarettes was associated with a higher adjusted odds of ever e-cigarette use (TV: AOR = 1.78, 95% CI: 1.46, 2.22; radio: AOR = 1.78, 95% CI: 1.46, 2.22). Bootstrap 95% CIs revealed significant indirect effects (TV: AOR = 0.90, 95% CI = 0.79, 0.96; radio: AOR = 0.87, 95% CI = 0.75, 0.98). Findings suggested that the relationship between TV/radio advertising exposure and ever e-cigarette use was mediated through e-cigarette harm perception. Of those who had never used an e-cigarette, advertising exposure via TV or radio was not associated with e-cigarette harm perception (TV: AOR = 0.74, 95% CI: 0.55, 1.02; radio: AOR = 0.67, 95% CI: 0.43, 1.06). Bootstrap 95% CIs indicated insignificant indirect effects (TV: AOR = 0.98, 95% CI = 0.96, 1.00; radio: AOR = 0.98, 95% CI = 0.95, 1.00). Thus, we did not detect the mediating effect of e-cigarette harm perception in the association between TV/radio advertising exposure and e-cigarette use susceptibility. 

E-cigarette advertising exposure through print media or large signs was not associated with e-cigarette harm perception, and bootstrap 95% CIs showed insignificant indirect effects. Hence, the mediating effect of e-cigarette harm perception was not observed in the association between advertising exposure via print media or large signs and outcomes. However, perceiving e-cigarettes as less harmful than cigarettes was found to be consistently associated with higher adjusted odds of e-cigarette ever use and susceptibility. 

## 4. Discussion

This study adds to the literature on the mechanism through which e-cigarette advertising exerts an effect on e-cigarette use behaviors among young adults. Participants reported relatively low levels of e-cigarette advertising exposure compared to those reported in prior studies [14,20]. The majority (59.4%) of our participants believed that e-cigarettes were equally or more harmful than cigarettes. This finding does not align with early research among college students showing most students perceive e-cigarettes as less harmful than cigarettes [14,20]. In fact, there are increasingly more people perceiving e-cigarettes to be equally or more harmful than cigarettes [33,34,35]. This may be related to the increasing media coverage on the serious injuries and health problems associated with e-cigarette use [36]. 

Our findings imply that the effect of e-cigarette advertising exposure on harm perception differs according to the source of advertising. Exposure to online advertising is associated with a low harm perception of e-cigarettes relative to cigarettes. In contrast, TV and radio exposure have an opposite association with such belief, whereas exposure to print media or large sign advertising is not associated with e-cigarette harm perception. Our findings are consistent with results from the National Youth Tobacco Survey indicating that youth exposure to e-cigarette advertising via retail stores and the Internet are more likely to perceive e-cigarettes as less harmful than cigarettes, whereas those exposed to TV advertising are less likely to believe so, and newspaper/magazine exposure is not associated with the harm perception [37]. The varied effects of advertising exposure on harm belief may be interpreted to be caused by the differing content by source since advertising is regulated differently for traditional sources (e.g., TV) and the Internet. Pokhrel et al. [15] concluded that social enhancement advertising (which projected e-cigarettes as socially appealing and fashionable), rather than health advertising (which emphasized the health benefits and harm reduction), is associated with explicit attitudes toward e-cigarettes as less harmful than cigarettes among young adults. Current research assessing the content of e-cigarette advertising has primarily focused on the Internet. E-cigarette advertising on the Internet typically emphasizes social enhancement and harm reduction [5,8]. Whereas for other sources, including TV and radio, advertising has rarely been studied. Therefore, not much is known about the content of e-cigarette advertising in sources other than the Internet. Future studies are needed to assess e-cigarette advertising content in a wide range of sources, explore how content would differ by source, and how the difference may affect people’s perceptions and attitudes toward e-cigarettes. 

Consistent with prior research among college students [38], our findings affirm that perceiving e-cigarettes as less harmful than cigarettes is associated with e-cigarette use and susceptibility. We further extend existing research by identifying the mediating effect of e-cigarette harm perception in the relationship between advertising exposure (via online, TV, and radio) and ever e-cigarette use, and between online advertising exposure and susceptibility to e-cigarette use. To date, not much research has been conducted on the mechanism of advertising exposure affecting e-cigarette use or related behaviors. In a study of non-smoking young adults, exposure to e-cigarette advertising was shown to be associated with a greater susceptibility to e-cigarette use among never users, and such association was mediated by explicit harm perceptions that e-cigarettes are less harmful than cigarettes [16]. This effect was observed for social enhancement advertising, but not for health advertising [16]. In the present study, we did not assess advertising content. Future studies are warranted to validate the findings through more rigorous research methods and to take into account advertising source and content. Longitudinal research needs to explore the possible causal mediating effect of e-cigarette perceptions on the relationship between advertising exposure and e-cigarette use. Research would also be desirable to identify more potential mediators, such as addiction perception of e-cigarettes and perceived social acceptability of e-cigarette use. 

This study has several limitations. First, causal inferences cannot be made based on our findings due to the cross-sectional nature of the study. Second, college students were recruited from one university and therefore, findings may not be generalizable outside of the study sample. Third, we used data collected from a larger study project that focused on a wide range of risky behaviors among college students; therefore, measures of e-cigarette advertising exposure were limited. Assessment of the frequency of advertising exposure, advertising exposure in more sources (e.g., retail stores), and advertising exposure in a more recent time point (e.g., past 30 days) would have been desirable. Fourth, because of the low frequency of current e-cigarette use, we did not examine the relationship between e-cigarette advertising exposure, harm perception, and current e-cigarette use. Fifth, self-reported data were subject to recall and reporting bias. Sixth, a mixed-mode data collection approach (online and telephone surveys) was implemented. However, we observed no difference in demographic characteristics by survey mode, indicating the approach was unlikely to create a significant bias. Lastly, our study was conducted before the outbreak of e-cigarette or vaping product use-related lung injury (EVALI) and the COVID-19 crisis. College students’ perceptions about e-cigarettes may have been changed during the EVALI outbreak or COVID-19 pandemic.

## 5. Conclusions

This study shows that the low harm perception of e-cigarettes relative to cigarettes mediates the association between e-cigarette advertising exposure (via online, TV, and radio) and ever e-cigarette use, and between online advertising exposure and e-cigarette use susceptibility among college students. Advertising exposure has a different impact on e-cigarette harm perception depending on the source of advertising exposure, but perceiving e-cigarettes as less harmful than cigarettes is consistently associated with e-cigarette use and susceptibility. Public health campaigns emphasizing the absolute harm as opposed to the relative harm of e-cigarettes may reduce the uptake of e-cigarettes among young adults. Regulatory actions should address e-cigarette marketing, particularly on the Internet. 

## Figures and Tables

**Figure 1 ijerph-19-06215-f001:**
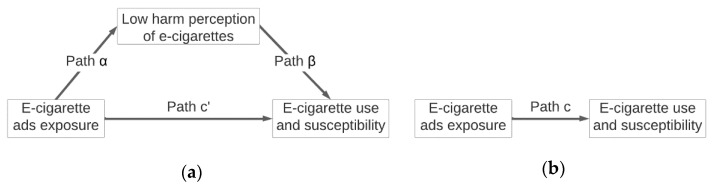
(**a**) Path diagram illustrating the mediating role of e-cigarette harm perception on the relationship between e-cigarette advertising exposure and e-cigarette ever use and susceptibility. (**b**) Total effect accounting for the relationship between e-cigarette advertising exposure and e-cigarette ever use and susceptibility.

**Table 1 ijerph-19-06215-t001:** Demographic and tobacco use characteristics of participants (*n* = 2078).

	*n*	(Weighted%)
Age		
18–20	549	(24.0)
21–24	811	(37.3)
25–34	549	(29.1)
≥35	169	(9.6)
Gender		
Male	785	(40.8)
Female	1280	(58.6)
Other	13	(0.6)
Race/ethnicity		
Non-Hispanic White	383	(19.7)
Non-Hispanic Black	353	(19.9)
Non-Hispanic Asian	575	(22.1)
Hispanic	671	(33.8)
Other	96	(4.5)
Native status		
US born	1289	(62.3)
Foreign-born	789	(37.7)
Cigarette smoking		
Never smokers	1542	(73.7)
Former smokers	396	(19.4)
Current smokers	140	(6.9)
Alternative tobacco use ^a^		
Never use	1380	(65.9)
Former use	539	(26.3)
Current use	159	(7.8)
E-cigarette use		
Ever use	351	(17.1)
Current use	73	(3.6)
Susceptibility to e-cigarette use ^b^		
Yes	308	(17.5)
No	1419	(82.5)
E-cigarette harm perception		
Less harmful than cigarettes	847	(40.6)
Equally/more harmful than cigarettes	1231	(59.4)
E-cigarette advertising exposure ^c^		
TV	406	(18.8)
Radio	167	(8.1)
Large signs	386	(17.9)
Print media	343	(16.1)
Online	650	(31.5)

^a^ Alternative tobacco products include cigars, hookah, little cigars, cigarillos, chewing tobacco, snuff or dip, and other tobacco products (non-cigarettes and non-e-cigarettes). ^b^ Among e-cigarette never users (*n* = 1727). ^c^ Multiple responses not adding up to 100%.

**Table 2 ijerph-19-06215-t002:** Mediating role of e-cigarette harm perception in the relationship between e-cigarette advertising exposure and e-cigarette ever use and susceptibility.

	Path α ^a^		Path β ^b^		Indirect Effect ^c^		Direct Effect ^d^		Total Effect ^e^	
	AOR	[95% CI]	AOR	[95% CI]	AOR	[95% CI]	AOR	[95% CI]	AOR	[95% CI]
Ever e-cigarette use (*n* = 2078)										
TV	0.70 *	[0.52, 0.91]	1.78 **	[1.46, 2.22]	0.90 *	[0.79, 0.96]	1.14	[0.75, 1.72]	1.06	[0.73, 1.51]
Radio	0.65 *	[0.43, 0.95]	1.78 **	[1.46, 2.22]	0.87 *	[0.75, 0.98]	1.06	[0.55, 1.82]	0.88	[0.51, 1.46]
Large signs	1.11	[0.83, 1.44]	1.78 **	[1.46, 2.22]	1.04	[0.95, 1.14]	1.08	[0.71, 1.60]	1.11	[0.78, 1.60]
In print media	1.09	[0.82, 1.49]	1.78 **	[1.44, 2.18]	1.04	[0.93, 1.14]	0.96	[0.60, 1.46]	0.98	[0.67, 1.44]
Online	1.29 *	[1.04, 1.60]	1.75 **	[1.44, 2.18]	1.08 *	[1.02, 1.18]	1.18	[0.80, 1.66]	1.22	[0.90, 1.66]
E-cigarette use susceptibility ^f^ (*n* = 1727)										
TV	0.74	[0.55, 1.02]	1.11 *	[1.08, 1.16]	0.98	[0.96, 1.00]	1.11 *	[1.02, 1.20]	1.31	[0.95, 1.82]
Radio	0.67	[0.43, 1.06]	1.11 *	[1.08, 1.16]	0.98	[0.95, 1.00]	1.02	[0.90, 1.16]	0.91	[0.55, 1.46]
Large signs	1.18	[0.88, 1.60]	1.11 *	[1.08, 1.16]	1.02	[0.98, 1.04]	1.00	[0.91, 1.09]	1.00	[0.71, 1.41]
In print media	1.16	[0.83, 1.60]	1.11 *	[1.08, 1.16]	1.02	[0.98, 1.04]	1.04	[0.95, 1.14]	1.11	[0.78, 1.57]
Online	1.31 *	[1.04, 1.72]	1.11 *	[1.08, 1.16]	1.02 *	[1.01, 1.04]	1.09 *	[1.02, 1.18]	1.60 *	[1.20, 2.06]

Notes. AOR: Adjusted odds ratio; CI: confidence interval. ^a^ Probit regression models assessing the association between advertising exposure and e-cigarette harm perception (mediator), adjusting for covariates (i.e., age group, gender, race/ethnicity, native status, cigarette smoking status, and alternative tobacco use). ^b^ Probit regression models assessing the association between mediator and outcomes (i.e., ever e-cigarette use, e-cigarette use susceptibility), adjusting for advertising exposure and covariates. ^c^ Indirect effect accounts for the effect of advertising exposure on outcomes through mediator. ^d^ Direct effect: Probit regression models assessing the association between advertising exposure and outcomes, adjusting for mediator and covariates. ^e^ Total effect: Probit regression models assessing the association between advertising exposure and outcomes, adjusting for covariates. ^f^ Among e-cigarette never users (*n* = 1727). * *p* < 0.05; ** *p* < 0.001.

## Data Availability

The data presented in this study are available on request from the corresponding author.

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
