# Peer review of "The Mediating Effect of E-Cigarette Harm Perception in the Relationship between E-Cigarette Advertising Exposure and E-Cigarette Use"

_ijerph, 2022, doi:10.3390/ijerph19106215_

Round 1

Reviewer 1 Report

Overall, I think the study presents an important topic and clear methodology. However, the measures used severely lacked detail and the authors were not convincing that the study contributes to our knowledge. Specifically:

-The description of tobacco use measures are inconsistent- I think never, ever and current should be the categories for cigarettes, e-cigarettes and other tobacco use.

-Given that less than 4% of the sample were current e-cigarette users, I question the representativeness

-I would have liked to know more about harm perceptions- were e-cigarettes perceived to be less harmful only when perceived harm of traditional cigarettes was very very high? Was the perceived harm due to the likelihood of becoming addicted to nicotine and ultimately smoking traditional cigarettes? We the harm, as the author suggest, due to potential contamination or product malfunction? These differ from believing the product itself, as normally used is more or less harmful.

-I think the authors need to provide much more reasoning as to why ad source matters. Additionally, knowing total time spent with each medium would help.

Reviewer 2 Report

This is an interesting study on a significant tobacco control issue.

However, some revisions are needed:

  1. The abstract section should be more informative. Please avoid obvious sentences such as "More studies are needed...". Please replate this with more informative conclusions or add more data to the results section (e.g., from table 2)
  2. The results section may be divided into sub-sections. This will be easier to follow.
  3. The discussion section is too short and limited. Please provide a more precise explanation of the obtained results and practical implications of this study.

Reviewer 3 Report

Thanks for the opportunity to review the manuscript by Nan Jiang and cols.

The authors show a study analyzing the mediating effect of e-cigarette harm perception in the above relationship among college students through a cross-sectional survey.

The manuscript looks well written, and the study design collected 2,112 college students (analytic sample: 2,078 participants).

The main results are really interesting. I have some minor comments to make.

In section 2.1. Survey procedures and participants, the authors state: Students who did not complete the online survey were invited to complete the survey via the phone. Please include a brief comment about the possible bias in this strategy.

Also, including a brief description of "Equally/more harmful than cigarettes," could grouping "equally and more harmful" have a dilutional effect in the response?

The authors include race/ethnicity among their items. Still, they do not make an analysis or description regarding the mediating effect of e-cigarette harm perception depending on race/ethnicity or gender status. Please include a brief comment in this regard.

The first paragraph's conclusion: -->This study contributes to the literature by identifying the low relative harm perception of e-cigarettes compared to cigarettes as a mediator in the association between e-cigarette advertising exposure (via online, TV, and radio) and ever e-cigarette use and/or susceptibility to e-cigarette use among college students.<-- In my opinion, it is not a conclusion; I think it would be better located as a final paragraph in the discussion, as a perspective.

Finally, according to the study design, this was done previous to the COVID-19 pandemic; this should be stated in the Methods section since in 2019 the EVALI was a great health concern and probably this condition could modify the e-cigarette harm perception. 

Round 2

Reviewer 1 Report

The measures used lack detail and the low number of current smokers makes me question the sample. The authors' response to this critique explained how this impacted the analysis, but my concern is the representativeness of the sample overall. Most importantly, the main finding in this study is that only online ad exposure leads to less harm perception. The lit review and discussion need MUCH more explanation as to the differences in content between sources of advertising. The authors note that little research has looked at TV e-cig ads. I think that if the authors performed a content analysis to pair with the current study it would be much more convincing.
